# A Survey on Sensor Failures in Autonomous Vehicles: Challenges and Solutions

**DOI:** 10.3390/s24165108

**Published:** 2024-08-07

**Authors:** Francisco Matos, Jorge Bernardino, João Durães, João Cunha

**Affiliations:** Polytechnic University of Coimbra, Rua da Misericórdia, Lagar dos Cortiços, S. Martinho do Bispo, 3045-093 Coimbra, Portugal; a21260249@isec.pt (F.M.); jorge@isec.pt (J.B.); jcunha@isec.pt (J.C.)

**Keywords:** autonomous vehicles, sensor failures, redundancy in AVs, fault tolerance

## Abstract

Autonomous vehicles (AVs) rely heavily on sensors to perceive their surrounding environment and then make decisions and act on them. However, these sensors have weaknesses, and are prone to failure, resulting in decision errors by vehicle controllers that pose significant challenges to their safe operation. To mitigate sensor failures, it is necessary to understand how they occur and how they affect the vehicle’s behavior so that fault-tolerant and fault-masking strategies can be applied. This survey covers 108 publications and presents an overview of the sensors used in AVs today, categorizes the sensor’s failures that can occur, such as radar interferences, ambiguities detection, or camera image failures, and provides an overview of mitigation strategies such as sensor fusion, redundancy, and sensor calibration. It also provides insights into research areas critical to improving safety in the autonomous vehicle industry, so that new or more in-depth research may emerge.

## 1. Introduction

Autonomous vehicles (AVs), or self-driving cars, are vehicles that perceive their surroundings and can move from point A to point B without human intervention. Through a combination of sensors, control theories, machine-learning algorithms, and computing, AVs can make real-time decisions. They are expected not only to reduce the number and severity of car accidents caused by human errors but also to reduce the carbon footprint through efficient driving [1]. However, despite these expected advantages, autonomous driving poses challenges such as a lack of governing legislation, a perceived increase in unemployment in the transportation sector, and cybersecurity threats [2,3].

Notwithstanding the current disbelief and uncertainty about fully autonomous vehicles [4,5], the number of vehicles using some degree of autonomous driving is increasing every year [6]. Many new vehicles now include driver assistance features that allow humans to perform multiple tasks on a daily basis, using sensors to perceive the vehicles’ surroundings and assist the human drivers in actions such as blind spot detection, lane change assistant, rear cross-traffic alert, forward cross-traffic alert, and adaptive cruise control, among many other examples. These actions represent incremental steps toward full autonomy. Nevertheless, current sensor technology is not reliable in all environments in which vehicles may operate [7,8]. Factors such as adverse weather conditions, complex urban settings, and sensor failures can significantly impair the performance of these systems. This unreliability poses challenges to the advancement of autonomous vehicles.

This survey aims to contribute to the development of solutions and strategies that increase the safety of autonomous vehicles by:Identifying and categorizing sensor failures.Identifying sensor limitations and the most frequent failures.Researching mitigation strategies for the categorized problems, such as sensor calibration, sensor fusion, radar ambiguity detection, and strategies for dealing with camera failures such as blur, condensation, broken lenses, heat, water, etc. Given that none of those mitigation strategies solves the issue of a complete failure of a sensor, we also explored how redundancy is applied in AVs.Discussing ongoing and potential future advances in the perception systems, such as continuous improvement of sensors, improving the robustness of machine--learning models, and the importance of redundancy in AVs.

Our research included surveying relevant papers. This survey highlights the importance of sensor fusion, redundancy, and continuous improvement in sensor calibration techniques to improve the safety and efficiency of AVs. It also shows that the topic of sensors in autonomous vehicles is becoming increasingly relevant, as the number of published papers per year has been increasing in recent years. Figure 1 shows the number of papers analyzed per year.

The remainder of this paper is organized as follows. Section 2 presents background concepts, introduces the architecture of an autonomous vehicle, and describes which sensors are used. Section 3 provides an overview of each sensor type, detailing its use and technical specifications. Section 4 describes mitigation strategies. Section 5 presents some ideas for future research in AVs. Finally, Section 6 concludes the paper by summarizing the findings.

## 2. Background Concepts

According to the J3016 standard proposed by the Society of Automotive Engineers (SAE) [9], autonomous vehicles can be classified into 6 levels of increasing degrees of autonomy. Level 0 refers to the weakest automation, where the driver has full control of the vehicle; Level 5 is the highest, corresponding to full automation, where the vehicle controls all aspects of driving and requires no human intervention. There are other classification schemes (e.g., the German Federal Highway Research Institute (BASt) and NHTSA [10]), but the J3016 is the most used in the industry and academia. The levels are shown in Figure 2.

These automation levels are important because they serve as general guidelines for how technologically advanced a car needs to be. Perception is a critical layer of the ADS (autonomous driving system), which uses sensors to perceive the environment and control the vehicle. Four main phases can be identified between sensing and controlling the vehicle: perception, planning and decision-making, motion and vehicle control, and system supervision [11]. This is illustrated in Figure 3. Perception of the surrounding environment is required at all levels of autonomy. Level 1 is the level that requires the least perception capability, as it only requires the perception needed to assist the driver in tasks such as parking and blind spot detection.

The main goal of the perception phase as described in [11] is to receive data from sensors and other sources (vehicle sensors configuration, map databases, etc.) and generate a representation of the vehicle state and a world model.

Sensors can be categorized into internal state sensors (or proprioceptive sensors), and external state sensors (or exteroceptive sensors) [12]. Proprioceptive sensors are those used to measure the (internal) state of the vehicle. This category includes sensors such as global navigation satellite systems (GNSS), inertial measurement units (IMUs), inertial navigation systems (INS), and encoders, which are devices used to provide feedback on the position, speed, and rotation of moving parts within the vehicle such as the steering wheel, motor, brake, accelerator pedals, etc. These sensors are used to obtain information about the vehicle’s position, motion, and odometry [13].

The autonomous vehicle (AV) can be positioned using relative or absolute methods. Relative positioning of an AV involves determining the vehicle’s coordinates based on (relative to) its surrounding landmarks, while absolute positioning involves determining the vehicle’s coordinates using a global reference frame (world) [14].

Exteroceptive sensors monitor the vehicle’s surroundings to obtain data on the terrain, the environment, and external objects. These sensors include cameras, LiDARs (light detection and ranging), radars (radio detection and ranging), ultrasonic sensors, and, in recent years, synthetic aperture radar (SAR). Although SAR is currently not implemented by manufacturers, we explored this sensor due to positive results in recent studies and its potential use in future vehicles.

In the typical AV use scenario, these sensors are used together to provide information about detection, lane occupancy, and more. Figure 4 shows an example of the type and location of sensors on a typical autonomous vehicle.

Sensors vary in technology and purpose, and each type of sensor has its weaknesses that are inherent in the technological capabilities that sensors use. To cover these weaknesses, some mitigation strategies are implemented, such as sensor calibration and sensor fusion [14]. Sensor fusion is a crucial part of autonomous driving systems [13,15] where input from multiple sensors is combined to reduce errors and overcome the limitations of individual sensors. Sensor fusion helps create a consistent and accurate representation of the environment in various harsh situations [16].

## 3. Sensors Used in the Perception Layer

This section provides an overview of each sensor type, detailing its use and technical specifications. Next, a categorization of sensor failures and limitations/weaknesses is presented, such as radar interference and harsh environments, offering a comprehensive overview of the various issues that can arise and their potential impact on AVs. This categorization is relevant to test teams, providing data that enhances the understanding of sensor issues in AVs to improve AV safety.

For the sake of clarity, we use the following definition from Avizienis, Laprie, Randell, and Landwehr [17]: *a **service failure** (or simply a **failure**) is an event that occurs when the delivered service deviates from correct service, either because it does not comply with the functional specification, or because this specification did not adequately describe the system function*. For example, an ultrasonic sensor **fails** when it delivers an incorrect service (wrong perception) due to interference between multiple sensors (which is called a fault). However, if this sensor does not perceive an object 100 m away, it is a **limitation**, but not a failure, since a maximum perception distance of 2 m is restrained by the functional specification. Since ultrasonic sensors are susceptible to too many service failures due to interference between multiple sensors, we call it a **weakness**.

### 3.1. Ultrasonic Sensors

Ultrasonic sensors use sound waves to perceive distance and detect the presence of objects. Ultrasonic sensors use sound waves in the frequency range of 20 to 40 kHz [18], which falls outside the human hearing range. The distance is calculated by emitting a sound wave and measuring its time-of-flight (ToF) until an echo signal is received. These sensors are directional with a very narrow beam detection range [18]. They are most used as parking sensors [19] and perform well in adverse weather conditions and dusty situations [7].

Given its narrow beam detection range, several ultrasonic sensors are needed to capture a full-field image. However, several sensors will interfere with each other and create interference, so a unique signature or identification code is required to reject echoes from other ultrasonic sensors in the vicinity. Ultrasonic sensors have a limited range, detecting obstacles up to 2 m [20].

### 3.2. RADAR: Radio Detection and Ranging

Radar sensors use millimeter waves (mm waves) and work by sending electromagnetic waves and then receiving them after they bounce back, using the Doppler shift effect. Radars can measure not only the exact distance but also the relative speed [15]. They operate at frequencies of 24/77/79 GHz, but most of the newly developed sensors operate in the frequency band of 76–81 GHz, as the use of the 24 GHz frequency was prohibited by regulators due to lower bandwidth, accuracy, and resolution. They typically have a perception range of 5 m up to 200 m [21], perform well in all weather conditions (rain, fog, and dark environments), and can accurately detect close-range targets on all sides of the vehicle [7,21]. Radars are used in blind spot detection (BSD), lane change assistant (LCA), rear cross traffic alert (RCTA), forward cross traffic alert (FCTA), adaptive cruise control, and radar video fusion.

One of the common problems of radars is that millimeter waves can give false positives because of possible bounced waves from the environment. Due to the increasing number of vehicles equipped with FMCW (frequency-modulated continuous-wave) radars, shared frequency interference is expected to become a problem [22].

### 3.3. LiDAR: Light Detection and Ranging

LiDAR sensors work by emitting pulses of light (laser) and measuring the time it takes for the light to bounce off objects and return to the sensor. By scanning the reflected laser beams emitted in various directions, LiDAR sensors can produce highly accurate spatial data, creating point clouds and distance maps of the environment [23]. LiDAR sensors are used to identify objects and pedestrians and avoid collisions. Due to their availability, 905 nm pulse LiDAR devices were used in the early AV systems. However, these 905 nm LiDAR systems have several important limitations, including high cost, inefficient mechanical scanning (in what concerns the movement necessary to direct the laser and sensor across its field of view), interference from other light sources, and eye-safety concerns leading to power restrictions that limited their detection range to approximately 100 m. This led to a shift to the retina-safe 1550 nm band, allowing higher pulse power with increased ranges of up to 300 m [23].

The performance of LiDAR in optimal environmental conditions is much better than that of radar, but as soon as there is fog, snow, or rain, its performance suffers [20,24]. In [25], the authors analyze the effects of mirror-like objects in LiDAR, explaining that laser scans can be completely reflected by mirrors, resulting in no range or intensity data, leading to the creation of a faulty map. They also explain the different behaviors of light reflection on multiple surfaces (Figure 5). This shows that LiDAR performance is strongly influenced by the environment.

### 3.4. Camera

Cameras can capture high-resolution details of objects up to 250 m away. Cameras are categorized into types that use visible (VIS) or infrared (IR) light, range-gated technology, polarization technology, and event detection [26]. They are based on one of two sensor technologies: charge-coupled device (CCD) or complementary metal-oxide-semiconductor (CMOS). CCD image sensors are produced using an expensive manufacturing process delivering sensors with high quantification efficiency and low noise. CMOS sensor technology was developed to minimize the cost of CCD fabrication at the expense of performance [11,14,21].

VIS cameras have a wide range of applications in AVs, including blind spot detection (BSD), lane change assistance (LCA), side view control, and accident recording. Deep learning algorithms can also be used with these cameras to detect and understand traffic signs and other objects [27,28,29].

IR cameras are passive sensors that use light in infrared wavelengths (780 nm to 1 mm), resulting in less light interference. The common application for AVs is in scenarios with illumination peaks, and in the detection of hot objects such as pedestrians [30,31], animals [32], or other vehicles [33].

Range-gated cameras are imaging systems that capture images based on the distance to the target by using a time-controlled gating mechanism. This gating mechanism synchronizes with a pulsed light source (such as a laser) to selectively allow light reflected from objects within a specific distance range to reach the camera sensor. By controlling the timing of the gate, the camera can effectively “slice” the scene at different distances, filtering out unwanted reflections and background noise to improve visibility in adverse conditions [34].

Polarization cameras are imaging devices that detect and measure the polarization state of light. Unlike conventional cameras that capture intensity and color, polarization cameras provide additional information about the angle and degree of light polarization. This extra layer of information can be used to infer various properties of objects and scenes that are not visible in standard images, and one of its main advantages is that it can detect transparent objects [35].

Event cameras are a type of vision sensor that capture changes in a scene asynchronously, as opposed to traditional cameras that capture full frames at fixed intervals. Event cameras detect and record individual pixel-level changes in brightness, called “events”, in real-time and with extremely low latency, and the main advantages are no motion blur and output over 1000 fps [36].

Cameras are strongly influenced by changes in lighting conditions, and by meteorological conditions such as rain, snow, and fog. Thus, cameras are typically paired with radar and/or LiDAR technologies to improve their resilience. In [37], the authors investigate the effects of degraded images on trained AI/ML agents, as seen in Figure 6. These camera failures were injected using the CARLA simulator [38], resulting in AV collisions, as shown in Figure 7, despite the application of some mitigation techniques (more in Section 4.6).

### 3.5. GNSS: Global Navigation Satellite Systems

The operating principle of the GNSS is based on the ability of the receiver to locate at least four satellites, and then compute the relative distance to each of them. Since the location of the satellites is known, the receiver can extrapolate its global position using trilateration.

GNSS signals are prone to many errors that reduce the accuracy of the system, including the following:Timing errors, due to variations between the satellite’s atomic clock and the receiver’s crystal clock.Signal delays, caused by propagation through the ionosphere and troposphere.Multipath effect.Satellite orbit uncertainties.

Current vehicle positioning systems improve their accuracy by combining GNSS signals with data from other vehicle sensors (e.g., inertial measurement units (IMU), LiDARs, radars, and cameras) to produce trustworthy position information [39,40,41]. This mitigation strategy is called sensor fusion and is discussed in Section 4.2. GNSS is susceptible to jamming, such as when the receiver encounters interference from other radio transmission sources. GNSS receivers also suffer from spoofing, when fake GNSS signals are intentionally transmitted to feed false position information and divert the target from their intended trajectory.

### 3.6. Inertial Measurements Units

Autonomous vehicles can detect slippage or lateral movement using inertial measurement units (IMU), which collect data from accelerometers, gyroscopes, and magnetometer sensors. Using this data, it is possible to detect the motion and orientation of the vehicle. The IMU, in conjunction with the other sensors, can correct errors and improve the sampling rate of the measurement system [21,41]. This approach is called inertial guidance.

### 3.7. Summary of Problems and Weaknesses of Interceptive and Exteroceptive Sensors

In Table 1, we present a comparison between each sensor’s advantages, as well as its limitations and weaknesses.

Table 2 presents, for each sensor type, the impact of the sensor failures on autonomous vehicles’ performance and safety.

## 4. Mitigation Strategies

This section analyzes mitigation strategies such as sensor calibration, sensor fusion techniques, radar interference, radar ambiguity detection, redundancy, camera image failures, and LiDAR mirror-like object detection.

### 4.1. Sensor Calibration

Subsequent processing steps such as sensor fusion, obstacle detection algorithms, localization, mapping, planning, and control require accurate sensor data. Thus, having calibrated sensors is of great importance. The following sections discuss three different forms of sensor calibration [14].

#### 4.1.1. Intrinsic Calibration

This type of calibration addresses sensor-specific parameters and occurs before extrinsic calibration and the execution of obstacle detection algorithms. Intrinsic calibration calculates the inherent parameters of a sensor, such as the focal lengths of a vision camera, to compensate for systematic or deterministic errors.

#### 4.1.2. Extrinsic Calibration

Extrinsic calibration is a Euclidean transformation that converts points from one 3D coordinate system to another. For example, it converts points from the 3D world or LiDAR coordinate system to the 3D camera coordinate system. This calibration calculates the position and orientation of the sensor relative to the three orthogonal axes of 3D space.

In [48], the authors propose an algorithm that calibrates the camera position using data provided by the IMU. In [49], extrinsic calibration between a camera and a LiDAR is achieved by using 3D calibration targets, and calibration of the sensors is then performed based on the extracted corresponding points of the object.

#### 4.1.3. Temporal Calibration

Temporal calibration establishes the synchronization rate of multiple sensor data streams. This is an important aspect because different sensors operate at different timings and with diverse latencies: For example, while a camera collects images at a given number of frames per second, a LiDAR or a radar may scan at a different rate [14]. This issue is very important because fusing data from non-synchronized sensors can induce system errors.

### 4.2. Sensor Fusion

Sensor fusion combines data originating from multiple types of sensors, taking advantage of their strengths while compensating for their weaknesses. For example, combining data from cameras and radar can provide high-resolution images and the relative speeds of obstacles in the area. Table 3 outlines the best sensor used in each factor [14].

There is a considerable amount of research on multi-sensor fusion [11,14,15,50,51,52,53,54,55,56,57,58,59,60]. The most common sensor combinations for obstacle detection are camera–LiDAR (CL), camera–radar (CR), and camera–LiDAR–radar (CLR). The CR sensor combination is the most used in multi-sensor fusion systems for environmental perception, followed by the CLR and CL [54,61]. A best combination of sensors does not exist, as the definition of “best” depends on functionality and price. The CR sensor combination produces high-resolution images and provides distance and velocity data about the surrounding obstacles [62,63,64,65]. Similarly, the CLR sensor combination improves resolution over longer distances and provides accurate environmental information using LiDAR point clouds and depth map data [14].

#### 4.2.1. Sensor Fusion Methodologies

Before fusing data, it is important to know which sensors to fuse and where to fuse—for instance, fuse only data corresponding to the front view of the vehicle or fuse everything around the car (bird’s eye)—and at which level the fusion should occur [63].

Multi-sensor data fusion (MSDF) frameworks have four levels: high-level fusion (HLF) also known as object-level; low-level fusion (LLF), also known as data-level fusion; mid-level fusion (MLF), also known as feature-level fusion; and hybrid-level fusion [66].

HLF (object-level fusion) approaches are often used due to their relatively low complexity when compared to LLF and MLF approaches. However, HLF provides insufficient information because classifications with lower confidence values are rejected, such as when there are multiple overlapping obstacles.

In contrast, the LLF (data-level fusion) technique integrates (or fuses) data from each sensor at the most basic level of abstraction (raw data). This preserves all information and can increase the accuracy of obstacle detection. This method requires precise external calibration, as it is heavily dependent on the good values of the sensors.

MLF (feature-level fusion) is an abstraction level between LLF and HLF; it combines multi-target features from the associated sensor data (raw measurements), such as color information from images or position features from radar and LiDAR, before performing recognition and classification on the merged multi-sensor features. However, MLF appears to be insufficient to achieve SAE automation levels 4 or 5 due to its limited sense of the environment and loss of contextual information [66].

Hybrid-level fusion can take the best of each level and merge the data. Although this gives good results, it greatly increases processing time [63].

#### 4.2.2. Sensor Fusion Techniques and Algorithms

Although sensor fusion methodologies and algorithms have been extensively researched, a new study [67] suggests that it remains a difficult process due to the interdisciplinary variety of the suggested algorithms. Another study [55] classified these techniques and algorithms as classical sensor fusion algorithms and deep learning sensor fusion algorithms. Classical sensor fusion algorithms, such as knowledge-based methods, statistical methods, probabilistic methods, and so on, use theories of uncertainty from data imperfections, including inaccuracy and uncertainty, to fuse sensor data, whereas deep learning sensor fusion algorithms generate various multi-layer neural networks that enable them to process raw data and extract features to perform challenging and intelligent tasks, such as object detection in an urban environment. In the field of AV, algorithms such as convolutional neural networks (CNN) and recurrent neural networks (RNN) are the most used perception systems, and other algorithms such as deep belief networks (DBN), and autoencoders (AE) [57] are also used.

Convolutional neural networks (CNNs) are specialized artificial neural networks designed to process and analyze visual data. They are particularly effective for tasks involving image recognition, classification, and computer vision.

Recurrent neural networks (RNNs) are designed to handle sequential data and temporal dependencies. They are commonly used for tasks involving time series data, natural language processing, and speech recognition.

Deep belief networks (DBNs) are a type of generative graphical model composed of multiple layers of stochastic, latent variables. They can learn to probabilistically reconstruct their inputs by stacking restricted Boltzmann machines (RBMs).

Autoencoders (AEs) are neural networks designed to learn efficient encoding of input data by learning to encode the input into a compressed representation and then decode it back to the original input. They are used for data denoising and image retrieval.

In [14] the authors show some additional approaches using different algorithms, and Table 4 summarizes these techniques.

### 4.3. Radar Interference Mitigation Strategies

As the number of automotive radar sensors in use increases, so does the likelihood of radar interference. According to the European MOSARIM (More Safety for All by Radar Interference Mitigation) project [71] to avoid interference, signals must differ in at least one dimension, such as time, frequency, location, or waveform. Radar mitigation techniques can be classified into four main categories [72].

Detection and suppression at the receiver.Interference is detected in the measurement data and eliminated by removing the corrupted data and recreating its value [73].Detection and avoidance.The radar actively modifies its signal to avoid interference in subsequent cycles when interference is detected in the measurement signal. This strategy is inspired by the interference avoidance mechanism of bats [74], which avoids interference rather than suppressing it locally.Interference-aware cognitive radar.The radar senses the entire operational spectrum and adaptively avoids interference using waveform modification [75].Centralized coordination.Self-driving cars are centrally coordinated to avoid radar interference [76]. Vehicles send their locations and routes to a control center, which models the radar operating schedule of vehicles in the same environment as a graph coloring problem and creates playbooks for each self-driving car to ensure that its radars work without interference.

### 4.4. Radar Ambiguity Detection Mitigation Strategies

The most important component of radar imagery is the reduction of angular ambiguities, which are signals caused by finite spatial sampling that can be misinterpreted as real targets [77].

A viable solution to the radar ambiguity problem in the automotive industry is to use a multichannel system [78] in which ambiguities are canceled if their angular distance from the real target is greater than the angular resolution of the multichannel array, as illustrated by the blue beam in Figure 8 [77,78].

### 4.5. Redundancy

Fully autonomous vehicles operate in real time and no matter the quality of the sensors or the robustness of an ADAS system once a sensor stops working it can put in danger the decision-making process of the system which can lead to accidents, which is why redundancy is implemented by car manufacturers to improve dependability. The concept of redundancy is duplication, triplication, etc., of one or more components of a system that perform the same function. For example, BMW’s autonomous driving system includes three AD (automatic driving) channels [79]. To avoid systematic faults and common cause failures, heterogeneous channels do not reuse hardware, software, algorithms, or sensors. In [80], the authors performed simulations with three AD channels operating simultaneously in the same test vehicle (Baidu Apollo 5.0, Autoware Auto AVP, and Comma.AI OpenPilot). They found that while the Apollo channel was usually the best, there were times when it failed while the other channels did not. In other words, the capabilities of some (less advanced) AD channels can complement those of other (more advanced) channels. This means that the combined capabilities of the multichannel system can potentially be greater than those of the most advanced channel alone. They then presented an architectural design pattern for cross-channel analysis, which is beyond the scope of this survey and will not be explored. Redundancy can be present in both hardware and software.

Hardware redundancy.This can include multiple components such as ECU, power supply, communication bus, input/output modules (sensors and actuators), communication module, etc. [81].Software redundancy.This involves having multiple autonomous driving systems [81].

### 4.6. Camera Image Failures Mitigation Strategies

The authors of [37] researched several mitigation strategies for camera failures such as those presented in Table 5.

In addition to these mitigation techniques, redundancy—which is also covered in this survey—is strongly encouraged: if a camera fails in a fully autonomous vehicle, then a redundant camera could help tolerate this failure. This also applies to other sensors such as LiDAR, radar, etc.

### 4.7. LiDAR Mirror-like Object Detection Using Unsupervised Learning

In addition to sensor fusion, unsupervised machine-learning algorithms can help solve this challenge. Clustering is suitable for AVs because it requires no learning and can be applied in real-time. DBSCAN (density-based spatial clustering of applications with noise) can categorize LiDAR data into reflective and non-reflective qualities based on their different characteristics [99].

The LiDAR scan results are then used as input for the DBSCAN clustering algorithm. Each outlier feature is classified according to range and intensity measurements. Diffuse surface qualities remain within normal ranges and intensities. Mirrors produce negative space qualities that result in no range and very low or no intensity measurements. Self-detecting qualities are characterized by extremely high-intensity measurements. Mirror reflection qualities often include a region with no range or intensity. These qualities can be used to remove clusters affected by mirror reflections [25].

## 5. Future Research Directions

Based on the literature review, we identified three main topics for future research, which are detailed in the following sections. These topics were selected based on our survey of current trends, and emerging areas of interest within the field. Our goal is to contribute to the community by pinpointing opportunities for advancing research.

### 5.1. Synthetic Aperture Radar (SAR) in AVs

SAR is a sophisticated radar technology that produces high-resolution images of the environment. While SAR sensors are typically used in aerospace applications, because of their unique characteristics, their use in autonomous vehicles (AVs) is being researched. The movement of the radar antenna over a target area creates a large aperture (hence the name) and provides high-resolution images [100]. Although radars are the best sensors to use against bad weather, they lack resolution in comparison to optical sensors and SAR can achieve finer spatial resolution by processing data from multiple antenna placements [101,102]. Some applications of this sensor are to detect pedestrians and other vehicles with far greater accuracy than currently existing radars [77].

The authors of [100] provide both a theoretical and experimental perspective on the role of SAR imaging in the automotive environment. Technological advances in array design (many antennas), analog-to-digital conversion, and increasing on-board computing resources allow the implementation of advanced signal processing algorithms in a power-efficient manner [103].

The authors of [104] propose a two-stage MIMO-SAR processing approach that reduces the computational load while maintaining image resolution. In [101], the effects of multipathing (detection of false targets due to multiple signal reflections) were analyzed, and it was concluded that the antenna layout has an impact on such reflections.

In [102], the authors use Radar in conjunction with SAR to improve the resolution of the radar by proposing a CSLAM (coherent simultaneous localization and mapping) model for unmanned vehicles, and they conclude that it provides better resolution and can be a good alternative to expensive LiDAR sensors.

According to the research, this type of sensor is new in AV. Its high resolution requires the development and testing of new algorithms and data processing methods in real-world environments.

### 5.2. Sensor Fusion Algorithms

When sensor weaknesses cannot be physically eliminated, algorithms (used to mitigate problems such as poor camera image, ambiguity, and radar interference) and sensor fusion play a central role. We also found that sensor fusion still needs work and existing models need more training.

Sensor fusion uses supervised algorithms to combine distinct types of sensors to compensate for sensor deficiencies. These algorithms require further development; however, the use of reinforcement learning paradigms in conjunction with supervised learning algorithms could aid in a sensor fusion scenario. In addition, reinforcement learning algorithms can be used to estimate the risk of failure of the sensor fusion solution early and allow for human intervention [8]. Machine learning models still require more training and data in challenging driving and weather situations. ML models are overly focused on what the vehicle feels. Human driving decisions also depend on assumptions about what other drivers will do (which is controversial because it can also be the cause of accidents). Some models are already investigating motion prediction, which is still in its early phases but promises excellent outcomes [105].

### 5.3. Advanced Driver Assistance Systems (ADAS) Redundancy

ADAS redundancy is also being explored and is a good area of research; it consists of using multiple control systems that can take over if the primary control system fails, ensuring that the vehicle can continue to operate safely even if the primary control system is compromised; these systems may or may not share sensors. A single system could use cameras and radar, and a redundant system could use only LiDAR. It is critical to explore this topic because as the level of autonomy increases, safety becomes a greater concern. Since redundancy increases costs, optimized systems are important for manufacturers [81].

## 6. Conclusions

This survey achieved several key objectives relevant to the field of autonomous vehicles (AVs), with a particular focus on the challenges and mitigation strategies associated with sensor failures. It provided an overview of the sensors currently used in AVs, categorized their problems, and explored the strategies implemented to mitigate these issues.

Several weaknesses are identified associated with different sensors used in AVs, such as ultrasonic sensors, radar, LiDAR, cameras, GNSS, and IMUs. This categorization is critical to understanding the vulnerabilities in the perception systems of AVs and forms the basis for developing more robust AV technologies. Techniques to address sensor failures such as sensor calibration, sensor fusion, redundancy, and specific failure mitigation strategies for cameras, LiDAR, and radar were analyzed. These strategies are essential for improving the reliability and safety of AVs.

Despite the benefits, the current mitigation strategies have limitations. For example, sensor fusion algorithms require extensive training and data, especially under challenging driving conditions. In addition, the high cost and complexity of advanced sensors like LiDAR, and the computational requirements of emerging technologies such as synthetic aperture radar (SAR) present significant challenges that can be explored.

In conclusion, this study provides valuable insights into the sensor technologies used in autonomous vehicles, identifies their vulnerabilities, and evaluates current and potential mitigation strategies, highlighting the importance of continuously improving sensor technologies and sensor fusion algorithms to achieve higher levels of autonomy and safety in AVs. By identifying and categorizing sensor failures, we have provided a clear understanding of the limitations and frequent issues faced by AV sensors. Furthermore, our exploration of mitigation strategies, including sensor calibration, sensor fusion, and redundancy, highlights the critical steps needed to improve AV safety and reliability. A discussion on ongoing and future advances in perception systems highlights the need for continuous sensor improvement, robust machine learning models, and the significance of redundancy.

We suggest that future research should focus on improving sensor fusion algorithms, particularly through the application of reinforcement learning and scenario-based training. Additionally, exploring the use of SAR in AVs and improving ADAS redundancy will be critical to advancing AV safety and performance.

## Figures and Tables

**Figure 1 sensors-24-05108-f001:**
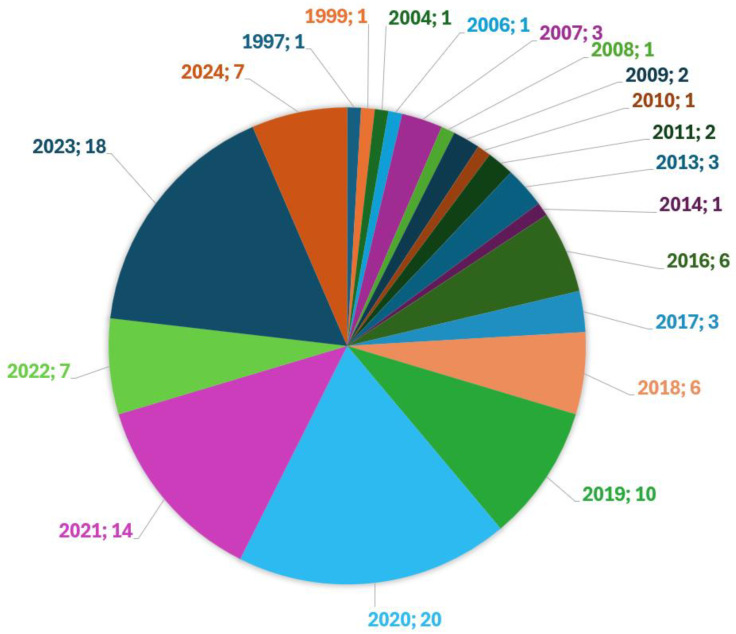
Number of papers reviewed per year (year; count).

**Figure 2 sensors-24-05108-f002:**
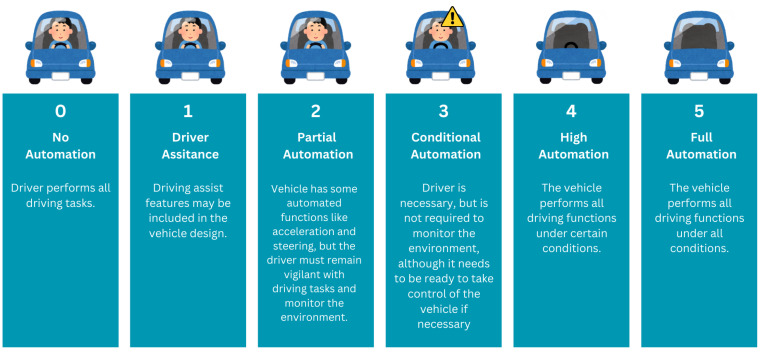
SAE automation levels.

**Figure 3 sensors-24-05108-f003:**
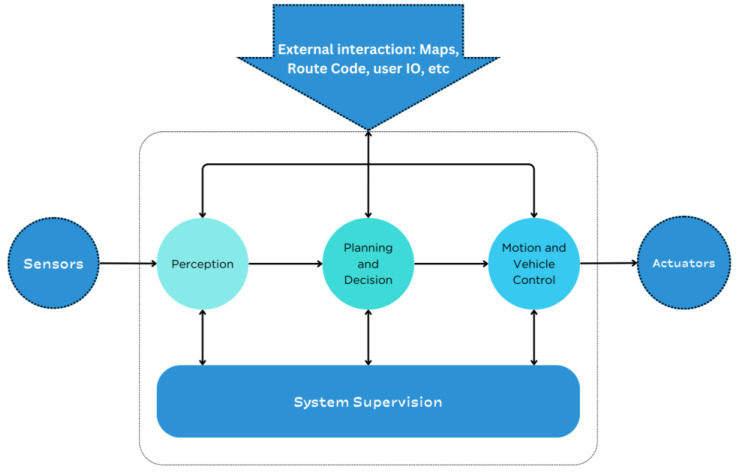
Functional architecture for an autonomous driving system.

**Figure 4 sensors-24-05108-f004:**
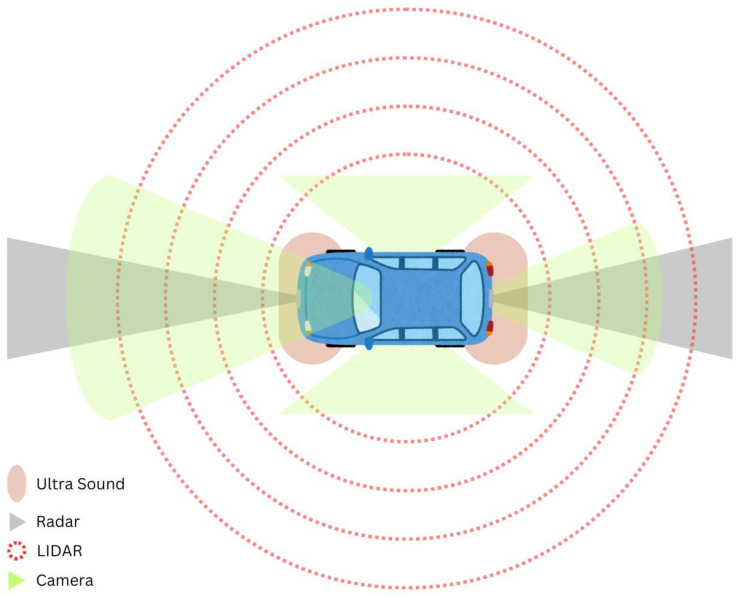
Sensors in an autonomous vehicle.

**Figure 5 sensors-24-05108-f005:**
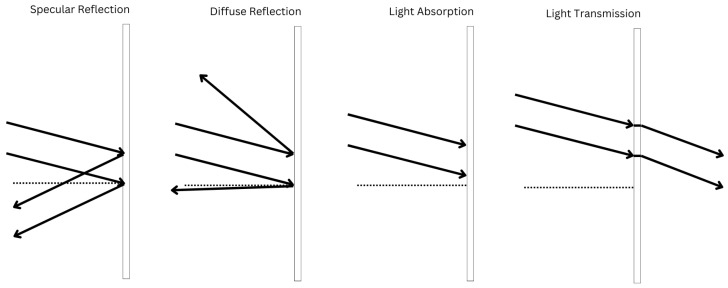
Light behavior on different surfaces.

**Figure 6 sensors-24-05108-f006:**
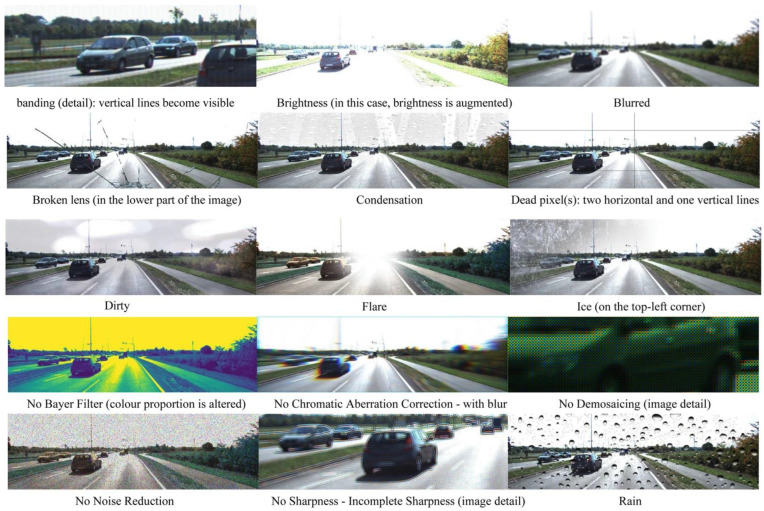
Simulated camera failures (from [37]).

**Figure 7 sensors-24-05108-f007:**
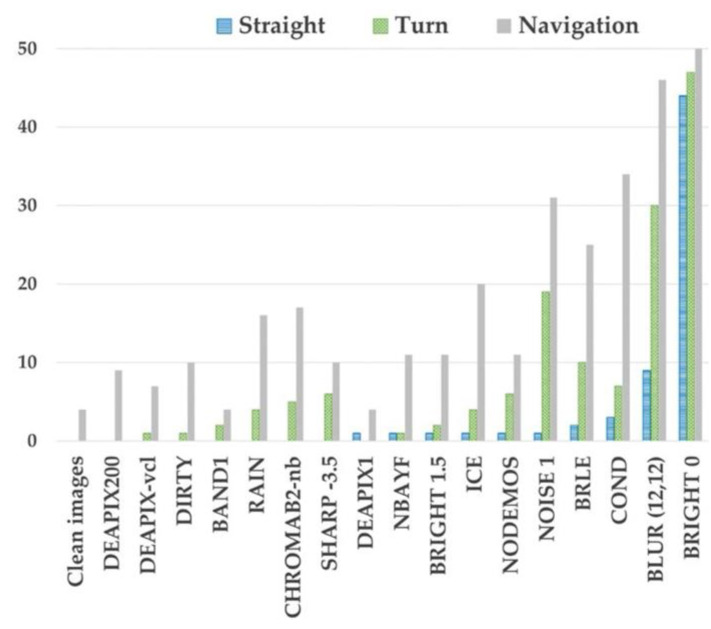
Number of collisions due to camera failures (from [37]).

**Figure 8 sensors-24-05108-f008:**
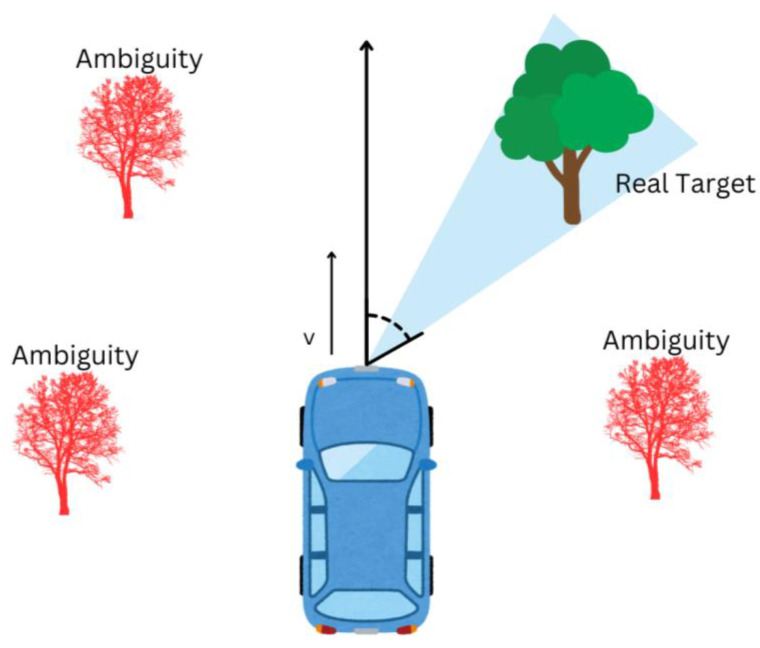
Radar ambiguities.

**Table 1 sensors-24-05108-t001:** Sensor’s advantages, limitations, and weaknesses (based on [21]).

Sensor	Advantages	Limitations/Weaknesses
Ultrasonic sensors	Affordable.	Limitations: Maximum range is 2 m. Very narrow beam detection range [18,42]. Weaknesses: Interference.
Radar	Long range (up to 200 m). Performs well in various weather conditions.	Limitations: Lower resolution when compared to cameras and LiDARs. Weaknesses: Many false positives. Radar interference [22].
LiDAR	High resolution. Range up to 200 m. Accurate distance measurement.	Limitations: Expensive. Weaknesses: Suffers extremely from the weather [20,24,43]. Reflective objects pose challenges [25].
Cameras	High resolution. Range up to 250 m. Capable of object recognition.	Limitations: Requires computational resources (Complex image processing). Weaknesses: Suffers from rough environmental conditions and several failures [37,43].
GNSS	Provides global positioning.	Limitations: Limited accuracy in certain conditions, such as dense areas [44]. Dependency on satellite visibility [45]. Weaknesses: Latency [44]. Vulnerable to signal jamming and spoofing [46].
IMU	Measures acceleration and rotation. Provides orientation information.	Limitations: Drift over time without external reference [47]. Weaknesses: Requires frequent calibration to maintain accuracy.

**Table 2 sensors-24-05108-t002:** Sensor failures and their impact.

Sensor Type	Failure	Impact
Ultrasonic Sensors	Wrong perception due to interference between multiple sensors.	Extreme range errors due to overlapping ultrasonic signals. Requires unique identification to reject false echoes.
Radar	False positives due to bounced waves.	This can lead to incorrect object detection or classification due to reflected signals from the environment.
	Wrong perception due to frequency interference from multiple radars.	Shared frequency interference may cause inaccuracies in object detection and tracking.
LiDAR	Detection performance degradation due to adverse weather conditions.	Reduced effectiveness in fog, rain, or snow, leading to incomplete or inaccurate spatial data.
	Missing or wrong perception due to reflection from mirrors or highly reflective surfaces.	This can result in faulty maps or missing data due to the laser beams being completely reflected.
Camera	Poor object detection due to variability in lighting conditions.	Performance can be significantly impaired in varying light conditions, leading to poor object detection.
	Image degradation due to rain, snow, or fog.	This can result in blurred or obscured images, affecting the accuracy of perception tasks.
	Misinterpretation in ADAS due to degraded images.	Degraded images can lead to AV collisions if the AI/ML systems fail to properly interpret the information.
GNSS	Timing errors due to clock differences.	This can affect the accuracy of location information, leading to incorrect positioning.
	Susceptibility to jamming and spoofing.	This can lead to loss of navigation accuracy or misdirection if the GNSS signals are blocked or falsified.
	Multipath effect and satellite orbit uncertainties.	This can lead to errors in location determination due to signal reflections and orbital inaccuracies.
IMU	Error accumulation and drift.	Errors in acceleration and rotational data can lead to inaccuracies in vehicle movement and orientation over time.

**Table 3 sensors-24-05108-t003:** Comparing sensors’ strengths in self-driving cars (based on [14]).

Factors	Best Sensor
Range	Radar
Resolution	Camera
Distance Accuracy	LiDAR
Velocity	Radar
Color Perception (e.g., traffic lights)	Camera
Object Detection	LiDAR
Object Classification	Camera
Lane Detection	Camera
Obstacle Edge Detection	Camera and LiDAR

**Table 4 sensors-24-05108-t004:** Sensor fusion algorithms and characteristics (based on [14]).

Sensor	Characteristics
YOLO	You Only Look Once (YOLO) is a single-stage detector that uses a single convolutional neural network (CNN) to predict bounding boxes and compute class probabilities and confidence scores for an image [68]. Advantages: Real-time detection (single-pass detection). Recognizes pedestrians and objects. Weaknesses: Lower accuracy than SSD. The system struggles to recognize dense barriers, such as flocks of birds, due to its limited ability to propose more than two bounding boxes. It has poor detection of small objects.
SSD	The Single-Shot Multibox Detector (SSD) is a single-stage CNN detector that converts bounding boxes into a collection of boxes with different sizes and aspect ratios to detect obstacles of various dimensions [69].Advantages: Real-time and accurate obstacle detection. Single pass. Detecting small objects can be challenging. However, it outperforms YOLO. Weaknesses: Poor feature extraction in shallow layers. Loses features in deep layers.
VoxelNet	VoxelNet is a generic 3D obstacle detection network that combines feature extraction and bounding box prediction into a single-stage, fully trainable deep network. It detects obstacles using a voxelized technique based on point cloud data [70].Advantages: No need for manual feature extraction. Voxelization improves LIDAR data management by reducing sparsity. Weaknesses: Training takes a large amount of data and memory.
PointNet	PointNet is a permutation-invariant deep neural network that learns global features from unordered point clouds using a two-stage detection approach [70].Advantages: Ability to maintain point clouds in any sequence, with permutation independence. Weaknesses: Difficult to generalize to unknown point configurations.

**Table 5 sensors-24-05108-t005:** Camera image failure mitigation strategies (based on [37]).

Component	Failure	Mitigations
Lens	Brightness	Brightness, if detected, can be compensated to some degree in post-processing. Images that are completely black or white are easy to detect, but recovering the original image is difficult.
	Blur	There exist several approaches for eliminating or correcting image blur [82,83]. For example, in [82], the authors define blind blur as a new joint optimization problem that optimizes the sparsity of both the blur kernel and the clear image.
	Broken Lens	Image processing can detect a broken lens, but reconstructing a clean image can be difficult.
	Dirty	Image processing software can remove localized rain and soil effects from a single image [84]. For example, physics-based approaches [85] can remove dust and dirt from digital photos and videos.
	Flare	Lens artifacts such as flare and ghosting can be minimized or eliminated from a single input image during post-processing to correct the image.
	Rain	To mitigate the effects of rain, we refer to Dirty’s statement and [84].
	Condensation	Several studies, including [86], address the issue of avoiding or eliminating condensation inside cameras.
Image Sensor	Banding	There are numerous strategies to reduce the visual effects of banding, such as using dithering patterns [87].
	Dead Pixel	Dead pixels can be detected using solutions that can be implemented on an embedded device [88].
	Spots	There are many approaches to eliminate spots based on image processing; for example, [89] identifies spots and computes their physical appearance (size, shape, position, and transparency) in an image based on camera parameters.
	Chromatic Aberration	Image processing can help to reduce chromatic aberration [90,91].
	Mosaic	Although efficient demosaicing methods exist [92,93] it is difficult to recover the image when demosaicing fails.
	Distortion	Lens distortion can be detected and measured [94] and corrected using image processing methods [95,96].
	Noise	There are several methods to reduce or eliminate noise and sharpening, including commercial tools. There are also methods available that work directly at the sensor level, e.g., [97,98].
	Sharpness	

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
