# Peer review of "A Survey on Sensor Failures in Autonomous Vehicles: Challenges and Solutions"

_sensors, 2024, doi:10.3390/s24165108_

Round 1
Reviewer 1 Report
Comments and Suggestions for Authors
Sensor Failures in Autonomous Vehicles: Challenges and Solutions
The manuscript presents an overview of the sensors used in Autonomous vehicles today, categorizes the sensor’s problems and failures that can occur, and provides an overview of mitigation strategies. I have some comments for the authors.
1. The introduction section of this manuscript fails to clearly state the unique contribution of this manuscript. It is recommended to provide a clear overview of the innovation and contribution of this manuscript on the basis of existing research in the introduction and conclusion sections.
2. Although the reference section covers a wide range, some of the literature is relatively outdated. Suggest citing more recent research findings to ensure the forefront of the manuscript. At the same time, check and standardize the reference format to ensure that it complies with the citation standards of the journal.
3. Some images in the current manuscript are not clear enough, such as Figures 6 and 7.
4. Please ensure that the tables in the manuscript are standardized, such as Table 5.
5. The current description of different types of sensors and their faults is relatively scattered. Suggest adding a systematic comparison table to summarize the advantages and disadvantages, common fault types, and their impacts of various sensors.
6. The depth of discussion on redundant design and sensor fusion technology is insufficient.
7. The outlook for future research directions in the manuscript lacks classification criteria, which will reduce the reading experience for readers.
8. There are some basic errors and unclear expressions in the manuscript. It is recommended to conduct comprehensive language proofreading to ensure consistency in the use of technical terms and accuracy in expression.
Author Response
We would like to thank the reviewer for their helpful and detailed comments. We expect to have addressed all the reviewer’s comments. The text of the paper is now clearer and substantially improved.
Below, we address the reviewer comments. In the new revised manuscript, the text modifications appear red (minor changes not mentioned by the reviewer, such as small grammar or vocabulary changes and rewritten sentences to make them clearer or better, were not highlighted.)
Sensor Failures in Autonomous Vehicles: Challenges and Solutions
The manuscript presents an overview of the sensors used in Autonomous vehicles today, categorizes the sensor’s problems and failures that can occur, and provides an overview of mitigation strategies. I have some comments for the authors.
Comment 1. The introduction section of this manuscript fails to clearly state the unique contribution of this manuscript. It is recommended to provide a clear overview of the innovation and contribution of this manuscript on the basis of existing research in the introduction and conclusion sections.
Response 1: The Introduction section was revised, and we added an explicit list of the main contributions of this manuscript (lines 38-53). We also made the abstract clearer about the main contributions of the paper (lines 10-13 and 16-18).
We also emphasized the main contributions of this manuscript in the conclusions, not only regarding the synthesis and categorization of sensors and how their failures can affect AVs, but also how new lines of investigation become necessary to make vehicles safer (lines 565-571).
Comment 2. Although the reference section covers a wide range, some of the literature is relatively outdated. Suggest citing more recent research findings to ensure the forefront of the manuscript. At the same time, check and standardize the reference format to ensure that it complies with the citation standards of the journal.
Response 2. We conducted new research using some new keywords, such as Proprioceptive sensors, IR Cameras, IR Cameras animal detection, IR Cameras vehicles detection, IR Cameras pedestrian detection, IMU Calibration, IMU-LiDAR calibration, Image Brightness Calibration and Image Blur Correction (see Table 1), and we have found new articles, mainly from 2024 (see lines 98-109), that we then used in this research to ensure that our manuscript reflects the latest research findings and remains at the forefront of the field. Additionally, we have created Figure 2 that shows the number of papers included in our survey per year, highlighting our efforts to incorporate up-to-date literature.
Furthermore, we have thoroughly checked and standardized the reference format to ensure full compliance with the citation standards of the journal.
Comment 3. Some images in the current manuscript are not clear enough, such as Figures 6 and 7.
Response 3. We decided to redo all the images. We just kept Figures 6 and 7 (now Figures 7 and 8) from the original paper, with explicit authorization form the authors.
Comment 4. Please ensure that the tables in the manuscript are standardized, such as Table 5.
Response 4. We have revised and standardized Table 5 (now Table 6) and Table 1.
Comment 5. The current description of different types of sensors and their faults is relatively scattered. Suggest adding a systematic comparison table to summarize the advantages and disadvantages, common fault types, and their impacts of various sensors.
Response 5. We agree with the reviewer’s comment, and in response we have made the following updates to our manuscript to provide a more systematic and comprehensive comparison:
- We updated Table 2 (page 12) to include the advantages of each sensor type.
- We have created a new table (Table 3, page 13) that specifically illustrates the impact of sensor failures on autonomous vehicle’s performance and safety.
We believe these additions will help to present the information in a more organized and accessible manner, enhancing the clarity and usefulness of our survey.
Comment 6. The depth of discussion on redundant design and sensor fusion technology is insufficient.
To address the reviewer’s concern, we have expanded the discussion on redundant design (lines 444-465) by adding additional information. Additionally, we have enriched our content related to sensor calibration (page 13, lines 315-319, 329-332), sensor fusion methodologies (page 14, lines 358-360; page 15, lines 378-379), and sensor fusion algorithms (page 15, lines 393-394, 401-408). We included references to recent articles that provide insights into deep learning algorithms relevant to sensor fusion (lines 383-395).
Comment 7. The outlook for future research directions in the manuscript lacks classification criteria, which will reduce the reading experience for readers.
Response 7. We acknowledge your concern about the lack of explicit classification criteria. We want to clarify that the future research directions were derived based on our review of the existing literature. Our approach involved recognizing emerging areas of interest within the field. Therefore, we clarify this in the introduction of section 6:
“Considering the literature review, we identified three main topics for future research, which are detailed in the following sections. These topics were selected based on our survey of current trends, and emerging areas of interest within the field.”
Comment 8. There are some basic errors and unclear expressions in the manuscript. It is recommended to conduct comprehensive language proofreading to ensure consistency in the use of technical terms and accuracy in expression.
Response 8. We have carefully reviewed and revised the manuscript to enhance clarity and readability. Some minor adjustments were made to grammar and vocabulary, along with rewriting certain sentences for improved clarity.
Reviewer 2 Report
Comments and Suggestions for Authors
This paper presents an overview of sensors used in autonomous vehicles, categorizing 11 types of sensor problems and failures, including radar interferences, detection ambiguities, and camera image problems. Additionally, it outlines mitigation strategies such as sensor fusion, redundancy, and sensor calibration. Overall, the survey is interesting, but the number of analyzed papers might be low, possibly due to the systematic approach taken. Additionally, I believe the fault mitigation topic, which is highlighted as a key feature of this literature review, is underdeveloped in the manuscript.
Some comments to improve the paper:
* As this is a systematic review, it is imperative to include this in the title and abstract of the manuscript. Also, include in the abstract the number of documents analyzed and the general description of the methodology used for the systematic review.
* I personally find systematic reviews useless because many papers relevant to the topic are excluded. It is then also important to explicitly state the objective of the literature review to analyze whether the searches are relevant to the objective of the systematic literature review.
* In general, the images in the document are of very poor quality. It is necessary to improve the definition of the photos, avoiding high compressions and preferably providing vectorized images for graphics and drawings. Particularly for figures 5, 6, and 7, although all figures need enhancement.
* Many figures include references in their captions. Is this because the figures contain information from these references but are not exact reproductions? Or are they reproduced exactly as they appear in the references? If it is the latter, ensure you have written permission to include them in your manuscript, as merely citing the reference is insufficient in this case.
* An aspect that is poorly addressed in the manuscript is sensor failures. Since this topic is specifically mentioned in the title, it is crucial to enhance the discussion on this aspect. Therefore, it is necessary to thoroughly address fault diagnosis issues, e.g., fault diagnosis observer for descriptor takagi-sugeno systems,neurocomputing; a review of convex approaches for control, observation and safety of linear parameter varying and takagi-sugeno systems, processes.
* Similarly, the future research direction section mentions sensor fusion to mitigate individual sensor errors and sensor redundancy to replace sensors in case of failure. However, it is essential to include diagnostic techniques for detecting faults, isolating them (identifying which sensor failed), and quantifying the faults.
* Fault-tolerant control strategies, which are implemented after a failure is diagnosed (detected, isolated, and quantified), could also be mentioned in a subsection of future research directions.
Comments on the Quality of English LanguageMinor editing of English language required
Author Response
We would like to thank the reviewer for their helpful and detailed comments. We expect to have addressed all the reviewer’s comments. The text of the paper is now clearer and substantially improved.
Below, we address the reviewer comments. In the new revised manuscript, the text modifications appear red (minor changes not mentioned by the reviewer, such as small grammar or vocabulary changes and rewritten sentences to make them clearer or better, were not highlighted.)
Comment 1. This paper presents an overview of sensors used in autonomous vehicles, categorizing 11 types of sensor problems and failures, including radar interferences, detection ambiguities, and camera image problems. Additionally, it outlines mitigation strategies such as sensor fusion, redundancy, and sensor calibration. Overall, the survey is interesting, but the number of analyzed papers might be low, possibly due to the systematic approach taken. Additionally, I believe the fault mitigation topic, which is highlighted as a key feature of this literature review, is underdeveloped in the manuscript.
Response 1. We would like to thank the reviewer for his valuable comments. Taking into consideration your comment about the number of papers, we have conducted a new search with additional keywords and added 42 new papers (Highlighted in the references section). To address the issue of fault mitigation, we included additional information concerning mitigation methodologies, such as, hybrid fusion (page 14, lines 378-379), sensor fusion algorithms (page 14, lines 393-394, 400-407), and redundancy (page 18, lines 444-448, 460-465).
Comment 2. As this is a systematic review, it is imperative to include this in the title and abstract of the manuscript. Also, include in the abstract the number of documents analyzed and the general description of the methodology used for the systematic review.
Response 2: We would like to clarify that our manuscript is a survey, not a systematic review. To reflect this accurately, we have followed the reviewers suggestion and updated the title to “A Survey on Sensor Failures in Autonomous Vehicles: Challenges and Solutions.” Additionally, we have revised the abstract to better describe the scope and approach of our survey, ensuring it accurately represents the nature of our work.
Comment 3. I personally find systematic reviews useless because many papers relevant to the topic are excluded. It is then also important to explicitly state the objective of the literature review to analyze whether the searches are relevant to the objective of the systematic literature review.
Response 3. We agree with your opinion regarding systematic reviews and acknowledge that they can sometimes exclude relevant papers. However, as mentioned in the previous comment we would like to clarify that our work is a survey, not a systematic review. We also revised our introduction to clearly state the objective of our survey (lines 38-53).
Comment 4. In general, the images in the document are of very poor quality. It is necessary to improve the definition of the photos, avoiding high compressions and preferably providing vectorized images for graphics and drawings. Particularly for figures 5, 6, and 7, although all figures need enhancement.
Response 4. We agree with your comment, and we have decided to redo all the images.
Comment 5. Many figures include references in their captions. Is this because the figures contain information from these references but are not exact reproductions? Or are they reproduced exactly as they appear in the references? If it is the latter, ensure you have written permission to include them in your manuscript, as merely citing the reference is insufficient in this case.
Response 5. We were able to get permission for image 6 and 7 (now 7 and 8). All the other images we decided to redo them all, because as the reviewer mentioned they did not have a good resolution.
Comment 6. An aspect that is poorly addressed in the manuscript is sensor failures. Since this topic is specifically mentioned in the title, it is crucial to enhance the discussion on this aspect.
Response 6. We thank you for pointing this out, we have added a table that describes failures per sensor type and the impact of these failures (Table 3, page 12). This table provides an overview, contributing to the understanding of the challenges associated with each sensor type and aligning with the specific focus mentioned in the title.
Comment 7. Similarly, the future research direction section mentions sensor fusion to mitigate individual sensor errors and sensor redundancy to replace sensors in case of failure. However, it is essential to include diagnostic techniques for detecting faults, isolating them (identifying which sensor failed), and quantifying the faults.
Response 7. We acknowledge the importance of diagnostic techniques for detecting faults. However, the aim of our survey is to categorize sensor failures, analyze their impact in AVs, and understanding what mitigation avenues are currently being researched.
Comment 8. Fault-tolerant control strategies, which are implemented after a failure is diagnosed (detected, isolated, and quantified), could also be mentioned in a subsection of future research directions.
Response 8. This suggestion is interesting, and we thank you. However, as in the case of the previous comment, fault tolerant strategies are not within the scope of our survey.
Round 2
Reviewer 2 Report
Comments and Suggestions for Authors
Many thanks to the authors for their efforts to improve their document based on my comments. However, I still consider that the manuscript has two major issues that must be solved:
According to Section 2, titled "Research Methodology," this paper is a systematic literature review. If this is not a systematic review, avoid detailing the article search method, including the keywords used and the articles' systematic selection/exclusion process. Limit your analysis to the article's topic, as it is unnecessary to include the number of articles excluded, the databases considered, and similar information. Otherwise, if it is a systematic review, ensure the title and abstract reflect this fact.
Additionally, the general topic of failures is still underdeveloped. Moreover, the concept of failure on a component is misunderstood. For instance, the authors consider that ultrasonic sensors fail when there is "wrong perception due to interference between multiple sensors." However, interferences are external factors and do not constitute a failure of the sensor itself. The measurement may be erroneous due to external noise or disturbances, but the sensor remains healthy. The same applies to external factors affecting cameras, such as fog or rain; the camera itself is healthy.
The manuscript has minor errors and typos, such as incorrect cross-references (e.g., line 98). Likewise, text similarity is high, and it needs to be decreased.
Comments on the Quality of English LanguageMinor editing of English language required
Author Response
We would like to thank the reviewer for their continued contribution in reviewing our manuscript and suggesting improvements. We expect to have addressed all the reviewer’s comments. Below, we describe how we addressed each of the reviewer’s comments. In the revised manuscript, the new text modifications appear in blue.
Comment 1: According to Section 2, titled "Research Methodology," this paper is a systematic literature review. If this is not a systematic review, avoid detailing the article search method, including the keywords used and the articles' systematic selection/exclusion process. Limit your analysis to the article's topic, as it is unnecessary to include the number of articles excluded, the databases considered, and similar information. Otherwise, if it is a systematic review, ensure the title and abstract reflect this fact.
Response 1: Thank you for your feedback, we appreciate your comment and agree with your point regarding the extensive detail in the Research Methodology section. To address this issue, we decided to reduce the extent of this information and remove details such as the number of articles excluded, keywords used, the organizational process, etc. We kept the Figure showing the number of articles per year as this information may be relevant to perceive the research trend on this topic. This resulted in the removal of section 2 and the addition of a brief explanation of our research process (lines 54-61).
Comment 2: Additionally, the general topic of failures is still underdeveloped. Moreover, the concept of failure on a component is misunderstood. For instance, the authors consider that ultrasonic sensors fail when there is "wrong perception due to interference between multiple sensors." However, interferences are external factors and do not constitute a failure of the sensor itself. The measurement may be erroneous due to external noise or disturbances, but the sensor remains healthy. The same applies to external factors affecting cameras, such as fog or rain; the camera itself is healthy.
Response 2: The reviewer points out an important issue in the manuscript, which does not clearly define the concept of “failure”, leading to misunderstanding from the reader. Therefore, we added lines 130-139 to clarify what “failure” means. This definition was taken from Avizienis, A., Laprie, J.-C., Randell, B., Landwehr, C.: Basic Concepts and Taxonomy of Dependable and Secure Computing. IEEE Trans. on Dep. and Sec. Comp. 1, 11–33 (2004), where “service failure“ (or simply a failure) is defined as an event that occurs when the delivered service deviates from the correct service, either because it does not comply with the functional specification, or because this specification did not adequately describe the system function.
We also clarified what “limitation” and “weakness” mean.
Therefore, if the functional specification of an ultrasonic sensor states that it should perceive an object within a certain range, not detecting this object due to external interferences is considered a failure.
We believe this addition to the manuscript make Table 1 and Table 2 (former Table 2 and Table 3) more understandable by the reader.
Comment 3: The manuscript has minor errors and typos, such as incorrect cross-references (e.g., line 98). Likewise, text similarity is high, and it needs to be decreased.
Response 3: We thoroughly revised the manuscript and corrected several typos and style errors. We are thankful to the reviewer for calling our attention to that issue. The changed text areas are noted in blue in the new manuscript version.
Regarding the text similarity, we are unsure about what it means. Therefore, we used tools available at our institution, including Urkund, which detects similarity across texts using a large database of published works. We confirmed that the similarity is primarily based on definitions and the list of references, which, by their nature, are naturally and inevitably similar to other works in the same field that use the same definitions and cite works we also cite. We believe that we cannot change this aspect as the definitions and references we use are essential to the manuscript.
Round 3
Reviewer 2 Report
Comments and Suggestions for Authors
I have no more comments